# Association of Echocardiographic Diastolic Dysfunction with Discordance of Invasive Intracoronary Pressure Indices

**DOI:** 10.3390/jcm10163670

**Published:** 2021-08-19

**Authors:** Hassan Tahir, James Livesay, Benjamin Fogelson, Raj Baljepally

**Affiliations:** 1Department of Cardiology, Heart Lung Vascular Institute, University of Tennessee Medical Center, Knoxville, TN 37920, USA; jlivesay@utmck.edu (J.L.); rbaljepa@utmck.edu (R.B.); 2Department of Internal Medicine, University of Tennessee Medical Center, Knoxville, TN 37920, USA; BAFogelson@utmck.edu

**Keywords:** physiological assessment, iFR, FFR, iFR/FFR discordance, diastolic dysfunction

## Abstract

Instantaneous wave-free ratio (iFR)-guided coronary revascularization has similar clinical outcomes compared to fractional flow reserve (FFR)-guided revascularization strategy. However, some studies have shown a discordance of around 20% between iFR and FFR. Although various factors have been reported in the literature to affect pressure indices and lead to such discordance, there is a paucity of data regarding the effect of diastolic dysfunction on functional assessment of coronary arteries. Our study aimed to investigate whether there was an association between echocardiographic left ventricular diastolic dysfunction and iFR/FFR discordance. This retrospective observational study evaluated 100 patients with angiographically intermediate coronary stenosis (50–70%) who underwent physiological testing with iFR and FFR. Transthoracic echocardiograms were reviewed to assess echocardiographic indices of diastolic function. The study population was divided into two groups based on diastolic function. iFR and FFR discordance was measured in each group and compared to evaluate the statistical difference. The mean age of the study population was 66.22 ± 10.02 years. Discordance between iFR and FFR was seen in 45.16% of patients with diastolic dysfunction compared to 24.64% of patients with normal diastolic function (*p* = 0.04). Multivariable logistic regression analysis indicated that echocardiographic E/e′ was independently associated with iFR/FFR discordance (*p* = 0.02). Left ventricular diastolic dysfunction is a significant factor that can lead to discordance between iFR and FFR and should be taken into account during coronary physiological testing.

## 1. Introduction

Functional assessment with instantaneous wave-free ratio (iFR) and fractional flow reserve (FFR) can help in the assessment of angiographically intermediate lesions in patients with stable ischemic heart disease and multivessel disease [1,2,3]. FFR correlates well with noninvasive stress tests regarding the functional significance of the coronary lesions [4]. However, FFR use has been limited by many factors in clinical practice, such as the use of vasodilator with its resultant side effects, cost, and procedure time [5]. iFR is a non-hyperemic pressure index that does not require vasodilators [6]. There has been an increase in the use of non-hyperemic pressure indices due to the ease of use and similar diagnostic accuracy as FFR [3,7]. Although various factors can lead to discordance between iFR and FFR, the effect of diastolic dysfunction on their discordance has not been studied before.

## 2. Materials and Methods

### 2.1. Study Design

We performed a single-center nonrandomized retrospective observational study. The study was approved by the institutional review board (IRB). The patient consent requirement was waived due to the retrospective nature of the study involving only chart review.

### 2.2. Study Population

Patients with angiographically intermediate stenosis (50–70%) who underwent functional assessment with both iFR and FFR were included in the study. Patients with both stable coronary artery disease (CAD) and acute coronary syndrome (ACS) were evaluated for functional assessment. In ACS patients, the functional assessment was performed in non-culprit vessels only. Exclusion criteria included severe valvular heart disease, restrictive cardiomyopathy, constrictive pericarditis, end-stage renal disease, contraindication to vasodilators, coronary artery bypass graft (CABG) patients, and inability to assess diastolic function due to poor-quality echocardiographic images. Discordance was defined as iFR ≥ 0.9 and FFR ≤ 0.80. Patients with iFR ≤ 0.89 were not included in the study as they underwent coronary revascularization without any further physiological testing. Patients who did not undergo a transthoracic echocardiogram before the procedure were excluded from the study.

### 2.3. Echocardiographic Evaluation of Left Ventricular Diastolic Function

Left ventricular diastolic function was evaluated by a transthoracic echocardiogram according to the 2016 American society of echocardiography (ASE) guidelines [8]. Most of the patients who had the iFR/FFR procedure underwent a transthoracic echocardiogram on the day of the procedure. Assessment of diastolic dysfunction included evaluation of septal/lateral tissue Doppler imaging (TDI) e′ velocity, E/e′ ratio, mitral valve E/A ratio, tricuspid regurgitation (TR) velocity, and left atrial volume index (LAVI) [8]. For tissue Doppler imaging (TDI) e′, data was acquired in 4 chamber view using pulse wave Doppler sample volume at septal and lateral basal annular regions. Wall filter and low signal gain were used to optimize spectral Doppler waveforms for accurate medial and lateral e′ measurement. Mitral valve (MV) peak E and A wave velocities were measured in apical 4 chamber view using pulsed wave sample volume placed between mitral leaflet tips. Mitral valve E/A ratio was calculated as MV E velocity divided by A wave velocity. Mitral E/e′ was calculated as MV E velocity divided by mitral annular average e′ velocity. Left atrium volume was measured in apical two chamber and four chamber views using the disc method, and indexed to the body surface area. Tricuspid regurgitation (TR) systolic jet velocity was measured in the parasternal view and four chamber views, and the highest velocity was measured with continuous-wave Doppler. The degree of diastolic dysfunction was graded based on the assessment of left ventricular filling pressures using an algorithmic approach described in ASE guidelines [8]. The echocardiographic evaluation also included assessment of left ventricular systolic function, valvular function, myocardial, and pericardial disease. The echocardiograms were reviewed by two independent readers. Patients with poor Doppler signals impeding the accurate measurement of echocardiographic parameters of diastolic function were excluded from the study.

### 2.4. iFR and FFR Physiological Assessment of Coronary Arteries

Patients underwent diagnostic coronary angiography via radial or femoral artery. Those patients with angiographically intermediate stenosis who underwent physiological assessment with both iFR and FFR were included in the study. Patients included were those with stable ischemic heart disease (ISHD) and acute coronary syndrome (ACS). In ACS patients, the functional assessment was performed in non-culprit vessels only. Philips Volcano pressure wire system was utilized for iFR and FFR measurement. Six French guide catheters were used for coronary engagement in all cases. The pressure sensor was advanced just distal to the guide tip and equalized to aortic pressure. Intracoronary nitroglycerin and intravenous heparin were given before advancing the pressure wire. The pressure wire was then advanced distal to the stenosis. The guide catheter was flushed with heparinized saline, and iFR was recorded. If any drift was noticed, the pressure sensor was pulled back to the tip of the guide catheter to recalibrate and reconfirm normalization to ensure the accuracy of the test. Patients with iFR ≤ 0.89 underwent revascularization of the lesion and were excluded from the study, while patients with iFR ≥ 0.90 underwent further assessment with FFR. Intravenous (IV) adenosine was used for hyperemia and was given at 140 mcg/kg/min for 2 to 3 min, and FFR was recorded. Coronary lesions with FFR ≤ 0.8 were considered hemodynamically significant.

### 2.5. Data Gathering and Statistical Analysis

Patients’ charts were reviewed to gather clinical history, demographic data, angiographic details, and iFR/FFR measurements. Echocardiograms were reviewed by two independent reviewers for assessment of diastolic dysfunction and its grading. Kolmogorov–Smirnov normality test was used to analyze the normal distribution of the data. Mean ± SD was used to express continuous variables, while proportion and percentages were used to express categorical variables. For continuous variables, an unpaired t-test was used to assess the statistical difference between the two groups. For categorical variables, the chi-squared test was used to assess the statistical difference. Multivariable logistic regression analysis was used to evaluate the association of various echocardiographic parameters of diastolic dysfunction with iFR/FFR discordance. Statistical significance was indicated by a two-tailed *p*-value of less than 0.05. Statistical analysis was performed using Graph Pad Prism (GraphPad Software Inc., San Diego, CA, USA), version 9.

## 3. Results

### 3.1. Baseline Characteristics

Table 1 summarizes the baseline characteristics of patients. A total of 100 patients were included in the study. The patients with normal diastolic function were assigned to one group (*n* = 69), and patients with diastolic dysfunction were assigned to the second group (*n* = 31). The mean age of the study population was 66.22 ± 10.02 years. The baseline characteristics of patients were similar between the two groups except for the increased prevalence of chronic systolic heart failure in patients with diastolic dysfunction. The most common clinical presentation was stable angina (70%). Out of 70 patients with stable CAD, 12 (17.14%) patients had chronic HFrEF while 7 patients (23.33%) out of a total of 30 patients with ACS had chronic HFrEF. In the diastolic dysfunction group, the majority of patients had grade 1 diastolic dysfunction (74.19%), followed by grade 2 (19.35%). A total of 26 out of 31 patients (83.87%) in the diastolic dysfunction group had elevated left ventricular diastolic pressure (≥15 mmHg). The most common coronary artery assessed for functional assessment was the left anterior descending artery (74%), followed by the right coronary artery (17%).

### 3.2. iFR/FFR Discordance

Discordance between iFR and FFR was seen in 31 patients. Comparison between the two groups showed that patients with diastolic dysfunction had significantly more iFR/FFR discordance compared with patients with normal diastolic function (45.16% vs. 24.64%, *p* = 0.04) (Figure 1). Echocardiographic parameters of left ventricular filling pressures, including E/e′ ratio, tricuspid regurgitation (TR) velocity, and left atrial volume index (LAVI), were assessed using multivariable logistic regression analysis. Of the three parameters, only MV E/e′ ratio was associated with iFR/FFR discordance (Odds Ratio, 2; 95% Confidence Interval, 1.56–2.63; *p* = 0.02) (Table 2). Of all the cases with iFR/FFR discordance, 80.7% were seen in the left anterior descending artery and 19.3% were seen in the right coronary artery. The correlation between LVEDP and E/e′ is shown in the scatter plot (Figure 2).

## 4. Discussion

FFR can help in the assessment of the functional significance of intermediate coronary stenosis in patients with stable ischemic heart artery disease who did not undergo prior stress testing [9]. In such patients, studies have shown better clinical outcomes with FFR-guided PCI compared to angiographic-guided PCI [10]. However, FFR has not been widely adopted, and its use in clinical practice has remained extremely low. This is mainly attributed to the mandatory need for hyperemic agents such as adenosine with their antecedent side effects, cost, and length of the procedure [5]. Non-hyperemic pressure indices such as iFR have multiple benefits that are appealing, including the ease of use, nonrequirement for vasodilators, and shorter procedural duration. Randomized control trials have shown that iFR-guided revascularization has similar clinical outcomes compared to FFR-guided PCI [11,12]. Despite the obvious advantages, there is a discordance of around 20% between iFR and FFR reported in multiple studies [13]. Some of the culprit factors reported in the literature leading to such discordance include age, beta-blocker use, heart rate, coronary stenosis degree, and location [7]. Elevated LVEDP is one of the earliest hemodynamic changes seen in diastolic dysfunction [14]. The effect of elevated LVEDP on coronary physiology has only been recently evaluated [15,16]. A recent study has shown that elevated LVEDP can lead to discordance between iFR and FFR [16]. There is a lack of data regarding the effect of echocardiographic diastolic dysfunction on invasive functional assessment of coronary arteries. Our study aimed to assess the effect of echocardiographic diastolic dysfunction on the discordance of iFR and FFR pressure indices.

Variation in coronary microvascular function may be the underlying mechanism leading to differences in the measurement of iFR and FFR, possibly because the non-hyperemic iFR index may be more affected by the microvascular function than the hyperemic FFR index. The effect of diastolic dysfunction on microvascular dysfunction has been previously reported in some studies. The PROMIS-HFpEF trial demonstrated a high prevalence of coronary microvascular dysfunction in HFpEF patients in the absence of macrovascular CAD [17]. Another study has described an association between microvascular dysfunction and echocardiographic parameters of diastolic dysfunction [18]. Kawata et al. found that coronary flow reserve (CFR) is associated with LV diastolic dysfunction in patients with type 2 diabetes [19]. However, there is a paucity of data regarding the effect of LV diastolic dysfunction on the discordance of pressure-flow indices. There are only a few studies that have evaluated the relation between diastolic dysfunction and coronary physiological testing. Tonre et al. concluded that iFR can be affected by LV diastolic dysfunction, and iFR may overestimate the severity of coronary stenosis in patients with LV diastolic dysfunction [20]. Another study evaluated the association between E/e′ and iFR/FFR and found that iFR was negatively correlated with E/e′, and no correlation was found between FFR and E/e′ [21]. The prognostic impact of iFR/FFR discordance on clinical outcomes has been evaluated in recent studies. Lee et al. studied the long-term outcomes of non-hyperemic pressure ratios (NHPRs) compared with FFR [22]. The authors concluded that deferred lesions with NHPRs and FFR discordance did not have adverse cardiovascular outcomes at five years compared to revascularized vessels [22]. Another study found that FFR-iFR discordance was not associated with an increased risk of composite end point of all-cause death, any myocardial infarction, and any revascularization among patients with deferred lesions at five years [23]. Large randomized controlled trials are needed to clarify the optimal treatment strategies for patients with lesions that have discordant iFR and FFR.

iFR is used as an alternative to FFR for physiological assessment of borderline lesions. Although some of the factors affecting the discordance of iFR and FFR have been previously evaluated, we report for the first time that echocardiographic diastolic dysfunction can impact invasive hemodynamic assessment with IFR and increase the discordance between iFR and FFR. In patients with echocardiographic diastolic dysfunction, the strategy of relying solely on iFR for hemodynamic assessment of coronary stenosis may not be the best option. As noted in our study, despite a negative iFR, additional physiological testing with FFR may be a reasonable approach to improve the diagnostic accuracy of coronary functional assessment. Due to the high prevalence of diastolic dysfunction in patients with coronary artery disease (CAD), it is imperative for interventional cardiologists to be cognizant that diastolic dysfunction can lead to the reduced diagnostic accuracy of non-hyperemic pressure indices and discordance between iFR and FFR.

## 5. Limitations

The main limitations of our study are the small sample size and non-randomization. In addition, there were few patients (9%) who did not undergo a transthoracic echocardiogram immediately prior to the coronary angiography procedure. Ideally, echocardiographic parameters of left ventricular filling pressures should have been assessed immediately prior to the procedure for all patients to accurately represent filling pressures during iFR/FFR measurement. In addition, patients with positive iFR and negative FFR discordance were not included in the study which may also be affected by diastolic dysfunction and elevated LVEDP.

## 6. Conclusions

Diastolic dysfunction is an important risk factor that can lead to discordance between iFR and FFR. Of the various echocardiographic parameters of diastolic dysfunction, mitral valve E/e′ was an independent predictor of discordance. A hybrid iFR/FFR strategy may be a reasonable approach and should be encouraged based on our study in patients with diastolic dysfunction to improve the diagnostic accuracy of coronary functional assessment. Further studies are needed to clarify the optimal treatment strategies for patients with lesions that have discordant iFR and FFR.

## Figures and Tables

**Figure 1 jcm-10-03670-f001:**
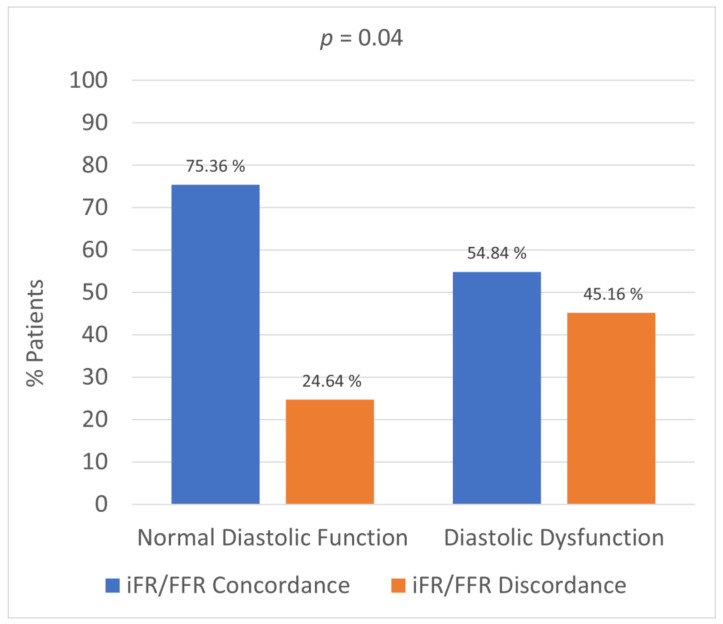
Comparison of iFR/FFR concordance and discordance between the two groups. iFR: instantaneous wave-free ratio; FFR: fractional flow reserve.

**Figure 2 jcm-10-03670-f002:**
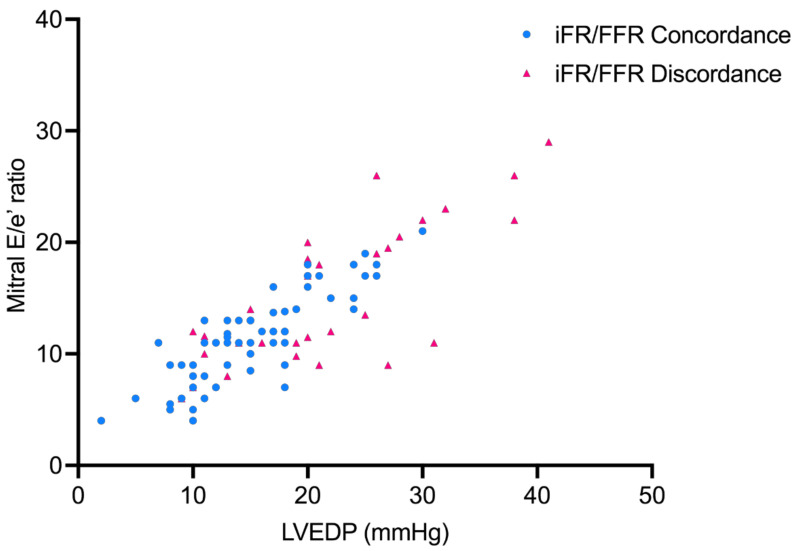
Scatter plot showing correlation between LVEDP and mitral E/e ratio in patients with iFR/FFR concordance and discordance. iFR: instantaneous wave-free ratio; FFR: fractional flow reserve; LVEDP: left ventricular end diastolic pressure.

**Table 1 jcm-10-03670-t001:** Baseline characteristics of patients.

Characteristics	All (*n* = 100)	Normal Diastolic Function (*n* = 69)	Diastolic Dysfunction (*n* = 31)	*p*-Value
Age (Yrs)	66.22 ± 10.02	65.68 ± 10.23	67.42 ± 9.43	0.43
Male	73 (73%)	52 (75.36%)	21 (67.74%)	0.42
Medical history				
Diabetes mellitus	33 (33%)	23 (33.33%)	10 (32.25%)	0.92
Hypertension	85 (85%)	57 (82.61%)	28 (90.32)	0.26
Hyperlipidemia	62 (62%)	46 (66.66%)	16 (51.61%)	0.15
Current Smoker	21 (21%)	11 (15.94%)	10 (32.26%)	0.06
Chronic HFrEF	19 (19%)	8 (11.59%)	11 (35.48%)	0.005
Chronic kidney disease	6 (6%)	3 (4.35%)	3 (9.68%)	0.29
Previous PCI	33 (33%)	19 (27.53%)	14 (45.16%)	0.08
Previous MI	16 (16%)	9 (13.04%)	7 (22.58%)	0.23
Previous CABG	6 (6%)	3 (4.34%)	3 (9.7%)	0.29
BMI (kg/m^2^)	30.46 ± 6.23	30.43 ± 6.41	30.53 ± 5.8	0.94
Echocardiographic findings				
Left ventricular ejection fraction (%)	53.47 ± 10.47	55.93 ± 7.69	49.03 ± 12.88	0.05
Diastolic dysfunction grades				
Grade 1	23 (23%)		23 (74.19%)	N/A
Grade 2	6 (6%)		6 (19.35%)	N/A
Grade 3	2 (2%)		2 (6.45%)	N/A
Clinical presentation				
Stable angina	70 (70%)	50 (72.46%)	20 (64.51%)	0.42
Acute coronary syndrome	30 (30%)	19 (27.54%)	11 (35.48%)	0.42
Pressure flow indices				
iFR	0.92 ± 0.02	0.92 ± 0.02	0.92 ± 0.02	0.97
FFR	0.84 ± 0.07	0.84 ± 0.06	0.83 ± 0.07	0.71
LVEDP (mmHg)	17.5 ± 7.08	14.1 ± 2.14	19.8 ± 7.21	0.05

Abbreviations: HFrEF, heart failure with reduced ejection fraction; PCI, percutaneous coronary intervention; MI, myocardial infarction; CABG, coronary artery bypass graft; BMI, body mass index; iFR, instantaneous wave-free ratio; FFR, fractional flow reserve; LVEDP, left ventricular end-diastolic pressure. Variables are expressed as proportions and percentages or mean ± standard deviation. *p* < 0.05 indicates statistical significance.

**Table 2 jcm-10-03670-t002:** Multivariable logistic regression analysis of echocardiographic left ventricular filling pressure parameters associated with iFR/FFR discordance.

Echocardiographic Parameters	Odds Ratio	95% Confidence Interval	*p*-Value
E/e′ > 14	2.0	1.56–2.63	0.02
LAVI > 34 mL/m^2^	0.88	0.08–7.58	0.91
TR velocity > 2.8 m/s	1.21	0.12–10.16	0.86

Abbreviations: LAVI, left atrial volume index; TR, tricuspid regurgitation.

## Data Availability

The data supporting the findings of this study are available from the corresponding author upon reasonable request.

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
