# Peer review of "Association of Echocardiographic Diastolic Dysfunction with Discordance of Invasive Intracoronary Pressure Indices"

_jcm, 2021, doi:10.3390/jcm10163670_

Round 1
Reviewer 1 Report
thank you
Reviewer 2 Report
the revised manuscript meets the requirements
This manuscript is a resubmission of an earlier submission. The following is a list of the peer review reports and author responses from that submission.
Round 1
Reviewer 1 Report
the research is interesting and in line with the ADVISE 2 trial results.
As you pointed out in the research discordance between iFR and FFR in patients with high LVEDP has already been demonstrated and it is possibly due to microvascular dysfunction.
E/e' ratio is commonly used in echocardiography to provide a non invasive LVEDP estimation. Therefore, if possible in clinical practice, it would be a useful piece of information that the imager can provide to the invasive cardiologist to choose the adequate technique for coronary stenosis assessment.
I did not understand If you used in the analysis e' avg values. I would have excluded from the analysis the patients who did not undergo echocardiography before the procedure.
Author Response
Response to Reviewer 1:
-The research is interesting and in line with the ADVISE 2 trial results. As you pointed out in the research discordance between iFR and FFR in patients with high LVEDP has already been demonstrated and it is possibly due to microvascular dysfunction. E/e' ratio is commonly used in echocardiography to provide a non invasive LVEDP estimation. Therefore, if possible in clinical practice, it would be a useful piece of information that the imager can provide to the invasive cardiologist to choose the adequate technique for coronary stenosis assessment.
Response: Thank you so much for the feedback.
-I did not understand If you used in the analysis e' avg values. I would have excluded from the analysis the patients who did not undergo echocardiography before the procedure.
Response: We used e’ average values in the study. None of the patients had echo after the procedure. There were few patients who had echo 2-3 days before the procedure rather than immediately before the procedure on the same day. Patients who did not undergo TTE before procedure were excluded from the study. I have added this statement to the exclusion criteria.
Reviewer 2 Report
In this manuscript, Tahir and colleagues investigated patients with a stenosis and a negative iFR result. They measured FFR in these patients and investigated whether echocardiographic parameters of diastolic dysfunction could clarify the discordance that arose between these patients. They found that patients with a negative iFR but a positive FFR more frequently had signs of diastolic dysfunction on echocardiography compared with patients with a negative iFR and negative FFR.
Comments:
- It would be of interest to study patients with positive iFR results as well, these patients are not studied for unclear reasons. I think the study would greatly benefit from enrolling these patients. Right now, we are left with the unaswered question how diastolic dysfunction impacts on patients with abnormal iFR, but normal FFR and patients with abnormal iFR and abnormal FFR. By excluding patients with abnormal iFR, I fear that selection bias influences the study greatly.
- Figure 1 is very unclear. No numbers are provided. It is unclear what the p-value represents.
- It is unclear what the present study adds to the previous study these authors published where they used invasively measured LVEDP instead of surrogate echocardiographic parameters. Please clarify.
- Please analyze whether the results were consistent in RCA vs. LCA lesions. Since it is known that LCA perfusion is predominantly diastolic, where a raised LVEDP is expected to have more impact, versus RCA perfusion where systolic flow also significantly contributes.
- In the discussion section, the authors describe that the study by Kawata et al. found that CFR was correlated with diastolic dysfunction. However, in the Justify CFR study (Petraco et al, Circulation Cardiovascular Interventions), it was found that iFR has a better correlation with CFR than FFR. How can the authors explain these results?
- In the disucssion section the authors state: iFR is considered as a surrogate to FFR for physiological assessment of borderline 180 lesions. This is untrue: iFR and FFR are different parameters with a different theoretical framework (de Waard GA et al., EHJ 2018).
Author Response
Response to Reviewer 2:
It would be of interest to study patients with positive iFR results as well, these patients are not studied for unclear reasons. I think the study would greatly benefit from enrolling these patients. Right now, we are left with the unanswered question how diastolic dysfunction impacts on patients with abnormal iFR, but normal FFR and patients with abnormal iFR and abnormal FFR. By excluding patients with abnormal iFR, I fear that selection bias influences the study greatly.
Response: Our study did not include patients with +iFR/-FFR discordance. Subsequent FFR testing in patients with +iFR was rarely performed in our institution. For that reason, we did not have patients to be included in the study. This certainly can lead to selection bias and is one of the limitations of the study. I have added these details to the limitations of our study.
Figure 1 is very unclear. No numbers are provided. It is unclear what the p-value represents.
Response: I have provided numbers for fig 1 to make it clearer and easy for interpretation.
It is unclear what the present study adds to the previous study these authors published where they used invasively measured LVEDP instead of surrogate echocardiographic parameters. Please clarify.
Response: Elevated LV filling pressures can be easily evaluated non-invasively by echocardiography. We recommended Hybrid iFR/FFR strategy in our previous study for patients with elevated LVEDP. Not all patients undergoing left heart catheterization undergo invasive LVEDP measurement for many reasons. It is this group of patients who can benefit from non-invasive echocardiographic assessment of LV filling pressures to guide iFR/FFR hybrid strategy.
Please analyze whether the results were consistent in RCA vs. LCA lesions. Since it is known that LCA perfusion is predominantly diastolic, where a raised LVEDP is expected to have more impact, versus RCA perfusion where systolic flow also significantly contributes
Response: The most common coronary artery assessed for functional assessment was left anterior descending artery (74%) followed by right coronary artery (17%). Of all the cases with iFR/FFR discordance, 80.7% were seen in left anterior descending artery and 19.3% were seen in right coronary artery. I have added these details to the manuscript.
In the discussion section, the authors describe that the study by Kawata et al. found that CFR was correlated with diastolic dysfunction. However, in the Justify CFR study (Petraco et al, Circulation Cardiovascular Interventions), it was found that iFR has a better correlation with CFR than FFR. How can the authors explain these results?
Response: Various studies have shown that Both FFR and CFVR have demonstrated to be useful to guide revascularization, with similar rates of major adverse cardiac events, and similar ability to detect myocardial ischemia in the presence of coronary stenoses. In the Justify CFR study, it was found that iFR has a better correlation with CFR than FFR. However, authors of the study stated that although pressure functional assessment (iFR or FFR) facilitates clinical application of physiology, it should not be seen as a biological equivalent of direct measurement of coronary flow. The authors also stated that when compared with FFR, iFR has a closer relationship with underlying CFVR should not be interpreted as superiority of one index over another. In studies of coronary physiology and ischemic heart disease, all intertest comparisons are limited by the lack of a true gold standard for the detection of myocardial ischemia.
In the disucssion section the authors state: iFR is considered as a surrogate to FFR for physiological assessment of borderline 180 lesions. This is untrue: iFR and FFR are different parameters with a different theoretical framework (de Waard GA et al., EHJ 2018).
Response: Thank you for the correction. I have changed the sentence to ‘iFR is used as an alternative to FFR for physiological assessment of borderline lesions.
Reviewer 3 Report
The relation between coronary perfusion pressure and non hyperemic coronary flow in patients has been extensively investigated in the past (Circ Cardiovasc Interv. 2014;7:301-311 and European Heart Journal 2016, 37, 2069–2080 ) and the use of resting pressure-flow indexes to detect the haemodynamic significance of coronary artery stenoses is suggested in CAD patients as a better option compared to hyperemic FFR. Discordance between non hyperemic indexes and hyperemic FFR in intermediate stenoses has been the target of several studies (Korean Circ J. 2018 Mar;48(3):179-190 https://doi.org/10.4070/kcj.2017.0393).
In this paper, Hassan Tahir et al. evaluated the contribution of LV diastolic function on this discordance – ranging from 10 to 30% - between vasodilator free instantaneous wave-free ratio (iFR) and fractional flow reserve (FFR) guided revascularisation in a small cohort of 100 CAD patients ( 70 stable CAD and 30 ACS) with intermediate coronary stenosis. The topic is of clinical interest and the authors demonstrate that an easy echocardiographic based stratification according to diastolic function prior to catheterisation helps identifying subjects more prone to develop a discordance between iFR and FFR : microvascular dysfunction and LVED pressure level, are the key determinants, especially in ACS, as the same authors demonstrated recently (Cardiol Res. 2021;12(2):117-125). Therefore, they suggest a hybrid approach, based on an echocardiographic parameter left ventricular filling pressure, i.e. the E/e’ ratio, to choose either iFR or FFR or both to guide revascularisation. Despite its potential clinical utility, there are some major concerns the authors must address in the Results and Discussion sections:
Although the echocardiographic assessment of diastolic function contributes to explain the discordance between the invasive indexes of functionally significant intermediate stenoses, its prognostic impact on iFR vs FFR guided revascularisation can be limited according to previous studies on larger patient cohorts, demonstrating that FFR–iFR discordance is not significantly related to an increased risk of adverse outcomes among patients with deferred lesions (J Am Heart Assoc. 2020;9:e016818. DOI: 10.1161/JAHA.120.016818 and J Am Coll Cardiol Intv 2019;12:2018–31) Further studies are needed to clarify the optimal treatment strategies for patients with lesions that have discordant iFR and FFR. This point should be clearly addressed in the Discussion and Conclusions sections.
The authors state that 26 out of 31 patients (83.87%) in the diastolic dysfunction group had elevated left ventricular diastolic pressure (>15 mmHg). How many of these patients were ACS? The relationship between E/e’ ratio as well as the other LV filling pressure indexes and LVEDP values in all 100 patients should be displayed, preferentially as a scatter plot, with distinct labelling of ACS and stable CAD cases. This additional information in the Result section would help to better understand the relation between the hemodynamic findings of their previous study (Cardiol Res. 2021;12(2):117-125), with the echocardioghraphic parameters measured here.
Although the clinical presentation is similar in the two groups of normal diastolic function and diastolic dysfunction, no information is available on the prevalence of chronic HF with reduced EF in the stable CAD compared to ACS patients. A chronically reduced EF is expected to be associated with microvascular dysfunction and increased LVEDP: would the results of the study be the same if the 19 patients with chronic HFrEF were excluded?
Author Response
Response to Reviewer 3:
Although the echocardiographic assessment of diastolic function contributes to explain the discordance between the invasive indexes of functionally significant intermediate stenoses, its prognostic impact on iFR vs FFR guided revascularisation can be limited according to previous studies on larger patient cohorts, demonstrating that FFR–iFR discordance is not significantly related to an increased risk of adverse outcomes among patients with deferred lesions (J Am Heart Assoc. 2020;9:e016818. DOI: 10.1161/JAHA.120.016818 and J Am Coll Cardiol Intv 2019;12:2018–31) Further studies are needed to clarify the optimal treatment strategies for patients with lesions that have discordant iFR and FFR. This point should be clearly addressed in the Discussion and Conclusions sections.
Response: Thank you for the feedback. I have added details in the manuscript.
The authors state that 26 out of 31 patients (83.87%) in the diastolic dysfunction group had elevated left ventricular diastolic pressure (>15 mmHg). How many of these patients were ACS?
Response: 35.48% of the patients in the diastolic dysfunction groups had ACS. Details added to the table.
The relationship between E/e’ ratio as well as the other LV filling pressure indexes and LVEDP values in all 100 patients should be displayed, preferentially as a scatter plot, with distinct labelling of ACS and stable CAD cases. This additional information in the Result section would help to better understand the relation between the hemodynamic findings of their previous study (Cardiol Res. 2021;12(2):117-125), with the echocardioghraphic parameters measured here.
Response: Thank you for the feedback. I have added scatter plot to the manuscript.
Although the clinical presentation is similar in the two groups of normal diastolic function and diastolic dysfunction, no information is available on the prevalence of chronic HF with reduced EF in the stable CAD compared to ACS patients. A chronically reduced EF is expected to be associated with microvascular dysfunction and increased LVEDP: would the results of the study be the same if the 19 patients with chronic HFrEF were excluded?
Response: Out of 70 patients with stable CAD, 12 (17.14%) patients had chronic HFrEF while 7 patients (23.33%) out of a total of 30 patients with ACS had chronic HFrEF. I have added details of prevalence of Chronic HFrEF in stable CAD vs ACS patients. It was not possible for us to exclude all HF patients and redo statistical testing due to small sample study.